Shifts in gut and vaginal microbiomes are associated with cancer recurrence time in women with ovarian cancer

Jacobson David 1 2
Moore Kathleen 3 Kathleen-Moore@ouhsc.edu
Gunderson Camille 3
Rowland Michelle 3 4
Austin Rita 1 2
Honap Tanvi Prasad 1 2
Xu Jiawu 5 6
http://orcid.org/0000-0002-4528-5877 Warinner Christina 7
http://orcid.org/0000-0002-4767-311X Sankaranarayanan Krithivasan 2 8
http://orcid.org/0000-0002-2198-3427 Lewis Jr Cecil M. 1 2 cmlewis@ou.edu
1 Department of Anthropology, University of Oklahoma , Norman, Oklahoma , United States
2 Laboratories of Molecular Anthropology and Microbiome Research (LMAMR), University of Oklahoma , Norman, Oklahoma , United States
3 Stephenson Cancer Center, University of Oklahoma Health Sciences Center , Oklahoma City, Oklahoma , United States
4 Saint Luke’s Hospital of Kansas City , Kansas City, Missouri , United States
5 Ragon Institute, MGH, MIT, and Harvard , Cambridge, Massachusetts , United States
6 Harvard Medical School, Harvard University , Boston, Massachusetts , United States
7 Department of Anthropology, Harvard University , Cambridge, Massachusetts , United States
8 Department of Microbiology and Plant Biology, University of Oklahoma , Norman, Oklahoma , United States
Lefkowitz Elliot
Electronic publication date: 2021 Jun 17
Publication date: 2021
Volume: 9
Electronic Location ID: e11574
Received 2020 Sep 18; Accepted 2021 May 18
Copyright: © 2021 Jacobson et al.
Copyright year: 2021
Copyright holder: Jacobson et al.
License: This is an open access article distributed under the terms of the Creative Commons Attribution License, which permits unrestricted use, distribution, reproduction and adaptation in any medium and for any purpose provided that it is properly attributed. For attribution, the original author(s), title, publication source (PeerJ) and either DOI or URL of the article must be cited.
License URL: https://creativecommons.org/licenses/by/4.0/

Keywords: Lactobacillus, Escherichia, 16S rRNA, Platinum-based chemotherapy

Funding: Stephenson Cancer Center at the University of Oklahoma Health Science Center National Institutes of Health NIH R01 GM089886 This work was primarily supported by seed funding from the Stephenson Cancer Center at the University of Oklahoma Health Science Center. National Institutes of Health (NIH R01 GM089886) provided support via shared chemical assay designs and reactions. The funders had no role in study design, data collection and analysis, decision to publish, or preparation of the manuscript.

==============================
Many studies investigating the human microbiome-cancer interface have focused on the gut microbiome and gastrointestinal cancers. Outside of human papillomavirus driving cervical cancer, little is known about the relationship between the vaginal microbiome and other gynecological cancers, such as ovarian cancer. In this retrospective study, we investigated the relationship between ovarian cancer, platinum-free interval (PFI) length, and vaginal and gut microbiomes. We observed that Lactobacillus-dominated vaginal communities were less common in women with ovarian cancer, as compared to existing datasets of similarly aged women without cancer. Primary platinum-resistance (PPR) disease is strongly associated with survivability under one year, and we found over one-third of patients with PPR (PFI < 6 months, n = 17) to have a vaginal microbiome dominated by Escherichia (>20% relative abundance), while only one platinum super-sensitive (PFI > 24 months, n = 23) patient had an Escherichia-dominated microbiome. Additionally, L. iners was associated with little, or no, gross residual disease, while other Lactobacillus species were dominant in women with >1 cm gross residual disease. In the gut microbiome, we found patients with PPR disease to have lower phylogenetic diversity than platinum-sensitive patients. The trends we observe in women with ovarian cancer and PPR disease, such as the absence of Lactobacillus and presence of Escherichia in the vaginal microbiome as well as low gut microbiome phylogenetic diversity have all been linked to other diseases and/or pro-inflammatory states, including bacterial vaginosis and autoimmune disorders. Future prospective studies are necessary to explore the translational potential and underlying mechanisms driving these associations.

Introduction

Ovarian cancer is the most deadly gynecological cancer (Siegel, Miller & Jemal, 2019); it kills approximately 14,000 women in the United States annually, accounting for 4.9% of all cancer-related deaths in females in the United States. In the majority of cases (>80%), ovarian cancer is not detected until stage III or later, primarily due to the nonspecific nature of ovarian cancer symptoms and lack of informative biomarkers (Torre et al., 2018; Jacobs & Menon, 2004). Early-stage (I or II) detection results in substantially greater 5-year survivability compared to late-stage diagnosis (III or IV): 70% versus 36% survival rate, respectively (Baldwin et al., 2012), highlighting the importance of discovering early-disease biomarkers.

The standard course of primary treatment in ovarian cancer is cytoreductive surgery (CRS) in combination with platinum-based chemotherapy (Raja, Chopra & Ledermann, 2012), which causes cytotoxicity through the formation of intra- and inter- strand adducts on DNA in cancer cells (Dasari & Tchounwou, 2014). The diameter of the remaining tumor after CRS, referred to as gross residual disease, is an important predictor of patient outcome, as individuals with no residual disease or residual disease <1 cm have improved survivability compared to those with tumors >1 cm after CRS (Chang et al., 2013). The combination of CRS and platinum-based chemotherapy is highly effective with approximately 80% of all patients showing no evidence of disease at the conclusion of initial therapy; however, recurrences occur in 70–80% of advanced stage patients and 20–25% of early-stage patients (Ushijima, 2010). Primary platinum resistance (recurrence of cancerous growth within six months of primary treatment cessation) develops in about 20% of patients, and is highly problematic because it is associated with a survivability of under one year and fewer effective treatment options (Davis, Tinker & Friedlander, 2014). Other patients may remain free of cancerous growth for more than two years, but the risk of recurrence and eventual development of treatment-resistant cancer is still unacceptably high (Bookman, 1999; Gore et al., 1990; Markman et al., 1991; Pfisterer & Ledermann, 2006). This merits a focus on discovering biomarkers of ovarian cancer and drivers of platinum-resistance to facilitate early cancer detection as well as better understand variation in treatment outcomes.

Recent evidence suggests that the human microbiome is an important factor in tumorigenesis, carcinogenesis, and effectiveness of chemotherapy (Bossuet-Greif et al., 2018; Iida et al., 2013; Perez-Chanona & Trinchieri, 2016; Schwabe & Jobin, 2013). Gut microbiome dysbiosis can influence colorectal carcinogenesis via production of genotoxic metabolites, such as colibactin, and through the promotion of a pro-inflammatory state, which contributes to cancer cell proliferation, angiogenesis, and metastasis (Bossuet-Greif et al., 2018; Schwabe & Jobin, 2013; Crusz & Balkwill, 2015; Zackular et al., 2013; Zackular et al., 2014). While most studies on the microbiome-cancer relationship have focused on the gut microbiome, there is growing evidence in support of the relationship between the vaginal microbiome and gynecological cancers. For example, human papillomavirus (HPV) is a known causative agent of cervical cancer (Champer et al., 2018; Chase et al., 2015; Muls et al., 2017), while pelvic inflammatory disease, which is associated with shifts in vaginal microbiome composition (Sharma et al., 2014), has been linked to ovarian cancer development (Lin et al., 2011). Yet, there are still many unknowns about the relationship between the vaginal microbiome and gynecological cancers (Champer et al., 2018; Chase et al., 2015; Muls et al., 2017; Xu et al., 2020). Likewise, links between the vaginal microbiome and platinum-sensitivity remain elusive; however, a previous study demonstrated that cancerous growths in mice with antibiotic-depleted gut microbiomes were less susceptible to platinum-chemotherapies compared to those with high gut microbiome diversity (Iida et al., 2013). The microbiome has a strong, bi-directional relationship with the host immune system (Shi et al., 2017; Thaiss et al., 2016) and immune cells in the mice with a depleted gut microbiome produced fewer reactive oxygen species (ROS) than control mice (Iida et al., 2013). Decreased ROS production and platinum-resistance in mice with low gut microbiome diversity suggests a role for the microbiome in response to platinum-chemotherapy because ROS play a part in cell apoptosis after exposure to platinum chemotherapeutic agents (Ozben, 2007).

In this study, we assessed how the vaginal and gut microbiomes vary in ovarian cancer patients with different platinum-sensitivities, with the aim of determining whether the human microbiome can be used as a biomarker of platinum-sensitivity.

Materials & methods

Study population

Patients, who carried a diagnosis of advanced (Stage III/IV) epithelial ovarian cancer and who were classified as primary platinum-resistant (platinum-free interval (PFI) from completion of primary platinum-based chemotherapy <6 months) or platinum super-sensitive (PFI > 24 months) and were being treated at the Stephenson Cancer Center at the University of Oklahoma Health Sciences Center, were approached to participate in this study. Patients were approached when they either developed primary platinum-resistant disease or when they were identified as platinum super-sensitive. Patients who had already been diagnosed with primary platinum-resistant disease or as platinum super-responders but had moved on to additional therapy, as well as patients on active anti-cancer therapy or in surveillance, were also included in this study. Patients were excluded if: (1) they were taking antibiotics at the time of sample collection or within 14 days prior to sample collection, or (2) they had active vaginal bleeding or known entero-vaginal fistulae. Samples were also collected from five individuals who were referred for ovarian cancer treatment at the Stephenson Cancer Center but ultimately had benign tumors; these served as a non-chemotherapy exposed control group. Sample collection periods and treatment procedures are outlined in Fig. S1. A brief summary of demographic and medical treatment history for participants in this study (n = 45, median age 62.2, age range 33–83) is provided in Table 1. This study was approved by the University of Oklahoma Health Sciences Center Institutional Review Board (February 22nd, 2016, reference #6458) and all participants gave written informed consent for their participation in the study.

Table 1 Study group demographics.

		Study group	
Clinical/lifestyle variable	PFI < 6 months (n = 17)	PFI > 24 months (n = 23)	Benign (n = 5)	
Median age (range)	64.6 (38.5–82.7)	63.6 (45.5–77.9)	57.8 (33.7–79)	
Self-reported ethnicity	White (% of study group)	15 (88.2%)	21 (91.3%)	3 (60%)	
African American (%)	1 (5.9%)	1 (4.3%)	0 (0%)	
Native American (%)	1 (5.9%)	1 (4.3%)	2 (40%)	
Cytoreductive surgery (CRS)	No surgery (% of study group)	1 (5.9%)	0 (0%)	4 (80%)	
Iterative CRS (%)&	13 (76.5%)	4 (17.4%)	1 (20%)	
Primary CRS (%)&	3 (17.6%)	19 (82.6%)	NA	
	Median months last platinum cycle (range)&	7 (1–38)	57 (8–150)	NA	
Expected estrogen positive (%)	4 (23.5%)	9 (39.1%)	1 (20%)	
Take probiotics	4 (23.5%)	4 (17.4%)	1 (20%)	
Vitamin supplement	6 (35.3%)	12 (52.2%)	1 (20%)	
Antibiotics within 6 months	14 (82.4%)	13 (56.5%)	3 (80%)	
Residual disease > 1 cm	6 (35.3%)	3 (13.0%)	NA	
Note:

Number of individuals belonging to each study group per clinical/lifestyle variable. The percent of individuals within each study category belonging to each metadata variable are presented in parentheses. Metadata variables that are significantly different between study groups are represented by & and Table S1 presents the p-value for each metadata variable.

All patients were treated initially with platinum and taxane chemotherapy for a planned six to eight cycles. These regimens included paclitaxel and carboplatin given every 21 days, paclitaxel given weekly with every 21st day carboplatin, or intraperitoneal administration of either cisplatin or carboplatin with intravenous and intraperitoneal paclitaxel. In patients with platinum resistant disease, standard of care options after recurrence included pegylated liposomal doxorubicin (PLD), weekly paclitaxel, gemcitabine, topotecan or bevacizumab given as monotherapy or in combination therapy. Patients were also screened for eligibility for clinical trials. Patients with a PFI > 24 months had not recurred at the time of study participation and were followed every 6 months with surveillance of Ca-125 values and exams. For those who had recurred beyond 24 months, treatment options included several platinum-based doublets including carboplatin and PLD given every 28 days, carboplatin and paclitaxel given every 21 days or carboplatin and gemcitabine given on a day one and day eight or day one and day 15 schedule. Each patient completed a minimum of six cycles of treatment and could undergo more cycles as long as the patient was responding and tolerating therapy.

Sample collection

Samples were collected during standard of care exams in the gynecologic oncology clinic at the Stephenson Cancer Center in Oklahoma City, OK. Catch-All Sample Collection Swabs (Epicentre) were used to collect vaginal and fecal samples. Vaginal swabs were collected from three sites per individual: vaginal introitus (VIT), mid-vagina (MDV), and posterior fornix (VPF), and then placed into a dry sample collection tube. Fecal samples were collected via a rectal digital exam, after which any stool collected was placed on a Catch-All swab and placed in a dry collection tube. Two swabs were collected from each site (bilaterally from the vaginal sites and sequentially for the rectal samples). Each participant completed a quality-of-life survey regarding their medical treatment history, antibiotic use within the past year, vitamin consumption, socioeconomic status, and other lifestyle metadata (Table S1).

Laboratory methods

DNA was extracted from the left-side vaginal swab and first fecal swab from each patient, using the MoBio PowerSoil DNA Isolation Kit (now Qiagen DNeasy PowerSoil Kit), following manufacturer’s protocols with the addition of a ten-minute incubation at 65 °C prior to the initial bead-beating step, as recommended in the Manual of Protocols for the Human Microbiome Project (McInnes & Cutting, 2010). A quantitative PCR (qPCR), using the SYBR Green PCR Master Mix (Applied Biosystems, Foster City, CA, USA) and primers targeting the V4 region of the bacterial 16S rRNA gene (Caporaso et al., 2011), was conducted; dilutions of Escherichia coli DNA corresponding to known 16S rRNA gene copy numbers were used as quantification standards for the DNA extracts. DNA extracts were amplified in triplicate, using Phusion High-Fidelity DNA polymerase (ThermoFisher Scientific, Waltham, MA, USA) and Illumina-compatible primers 515F and 806R (targeting the V4 region of the16S rRNA gene) with error-correcting Golay barcodes incorporated into the 806R reverse primer (Caporaso et al., 2011). PCR products were pooled in equimolar concentrations, purified with the MinElute PCR purification kit (Qiagen, Valencia, CA, USA), then size-selected between 300 and 450 basepairs using a PippenPrep, quantified using KAPA Biosystems Illumina library quantification kit, and sequenced across multiple runs of an Illumina MiSeq (500 cycles paired-end sequencing, v2 reagent kit).

Bioinformatic methods

AdapterRemoval (v2) (Schubert, Lindgreen & Orlando, 2016) was used to filter out reads with uncalled bases, reads with Phred quality threshold <30, and reads less than 150 bp in length. Quality filtered paired-end reads were merged using AdapterRemoval (v2) (Schubert, Lindgreen & Orlando, 2016) and then demultiplexed with QIIME (v1.9), followed by removal of chimeric sequences and low-abundance (<5 total sequences) reads (Caporaso et al., 2010). The remaining sequences were used for de novo Operational Taxonomic Unit (OTU) clustering with USEARCH (v10) at 97% sequence similarity (Edgar, 2010). Taxonomy was assigned to each OTU representative using the EzBioCloud 16S rRNA gene database (Yoon et al., 2017). The resulting OTU table was rarefied to 9000 reads and downstream analysis was performed in QIIME (v1.9) (Caporaso et al., 2010). The post-rarefaction sample breakdown was: PFI > 24 (n = 23), PFI < 6 (n = 17), and benign (n = 5). Further details of the bioinformatic methods are given in the Supplemental Material.

Statistical methods

Phylogenetic diversity and weighted/unweighted UniFrac (Lozupone & Knight, 2005) metrics were generated in QIIME (v1.9) with FastTree2 (Price, Dehal & Arkin, 2010). Tests for significance between study groups for alpha and beta diversity were performed using Kruskal-Wallis and PERMANOVA tests, respectively, in R (Team RC, 2013). Vaginal samples were classified into clusters by the dominant bacterial taxon found in each sample, as determined by Ward hierarchical clustering (Ward, 1963), and visualized as a heatmap using the gplots package in R (Warnes et al., 2016). If there were no dominant bacteria, the sample was classified as diverse. Median-unbiased estimated odds ratios were calculated to determine whether study groups had significantly different odds of dominant bacteria; reported odds ratios and 95% confidence intervals were log-transformed. Kruskal–Wallis tests with a Benjamini and Hochberg false-discovery rate adjustment were used to evaluate differential abundance of individual taxa between study groups. Odds ratios were calculated using epitools (Aragon, 2012) in R. Plots were generated using the ggplot2 (Wickham, 2016) and ColorBrewer (Harrower & Brewer, 2003) packages in R.

Results

Vaginal microbiome

Samples from the different vaginal sites (VIT, MDV, VIT) that originated from the same individual showed similar taxonomic beta-diversity profiles (Figs. S2A–S2C). Sequencing failed for at least one of the three vaginal sites of eleven individuals (Supplemental Material), which presented difficulty in analyzing each vaginal site individually as the failed specimens were not limited to only one vaginal site. Due to the similarity in composition between vaginal sites in each individual and the relatively high number of samples which failed sequencing, we concatenated sequencing reads from each individual’s three vaginal samples into a single representative vaginal microbiome sample per individual, and then performed downstream analysis with this single representative sample, unless otherwise noted. Combining vaginal sites for each individual allowed us to retain a sample size of 45. Firmicutes was the most dominant phylum in the vaginal microbiome and it was found at over 50% relative abundance in 40% of individuals (Fig. S3A), while Proteobacteria, Bacteroidetes, and Actinobacteria were the next most abundant phyla and found at >50% relative abundance in the vaginal microbiome of 13.3%, 13.3%, and 6.7% of individuals, respectively. At the genus level, Lactobacillus, Prevotella, Escherichia, Gardnerella, and Streptococcus were the most dominant bacteria and accounted for 57.2% of all reads (Fig. S3B).

Individuals were grouped into the following study groups as outlined in the methods section (Fig. S1): PFI < 6 Months (n = 17), PFI > 24 Months (n = 23), and benign (n = 5). Table 1 presents summary statistics for these study groups. Iterative cytoreductive surgery (iCRS) was more common in PFI < 6 months while primary cytoreductive surgery (pCRS) was more common in PFI > 24 months individuals (log Odds Ratio (OR) = 3.02, 95% CI [1.37–4.68], p-value = 0.0003). As expected, PFI < 6 months individuals underwent platinum chemotherapy treatment more recently than PFI > 24 months individuals (p-value = 4.32 × 10−7) but no other metadata variable was significantly associated with PFI status (Table S2).

Vaginal microbiome communities were clustered into five community-dominance groups using Ward hierarchical clustering: Lactobacillus cluster, Escherichia cluster, Gardnerella cluster, Prevotella cluster, and a high diversity cluster (Fig. 1). Results below are presented demonstrating how each vaginal community (excluding Gardnerella due to small cluster size) relate to clinical variables, p-values for each vaginal community and clinical variable are given in Table S3.

Figure 1 Heatmap of the 25 most abundant genera in the vaginal microbiome.

Each column represents a single individual’s vaginal microbiome, color coded by study group. Colors of each cell are based on a heatmap, ranging from 0 reads of the bacteria in that individual (white) to 9,000 reads of the bacteria in that individual (red). Samples were rarefied to 9,000 reads, so bright red indicates every read in the sample comes from that bacteria. Samples were clustered together based on similarity of vaginal microbiome using Ward hierarchical clustering. 11 microbiomes were Lactobacillus-dominated, six Escherichia-dominated, three Gardnerella-dominated, nine Prevotella, and 16 highly diverse.

Dominance groups were evenly distributed between patients with PFI < 6 months, PFI > 24 months, and benign, with the exception of higher-than-expected dominance of Escherichia in patients with PFI < 6 months; five of the six Escherichia-dominated vaginal communities identified with hierarchical clustering belonged to patients with PFI < 6 months (Fig. 1). Vaginal microbiomes dominated by Escherichia had higher odds of occurring in PFI < 6 months compared to PFI > 24 months (log OR = 2.812, 95% CI [0.267–5.62], p-value = 0.024, Fig. 2A). Additionally, one of the patients with PFI < 6 months and one of the patients with benign pathology identified with a ‘diverse’ vaginal microbiome had Escherichia at greater than 20% relative abundance, while no other patients with PFI > 24 months had Escherichia relative abundance above 5% (Fig. 1). In total, 35.3% of the patients with PFI < 6 months had Escherichia at greater than 20% relative abundance in the vaginal microbiome, compared to 4.34% of PFI > 24 months and only one of five benign individuals. Even though iterative cytoreductive surgery and Escherichia are more common in PFI < 6 months individuals, we found no significant relationship between iCRS and Escherichia abundance (Table S3, Fig. 2A, p-value = 0.292). Moreover, outside of the relationship between Escherichia and platinum resistance, Escherichia showed no significant association with any of the other health or lifestyle factors we examined (Table S3, Fig. 2A). Although Escherichia is a common lab-grown bacterium and found in feces, our analysis demonstrates Escherichia abundance in the vaginal samples is biological and not a technical artifact (Supplemental Material).

Figure 2 Log-transformed odds ratio vaginal microbiome dominance.

Log-transformed odds are represented by the orange circle and the bars represent 95% confidence intervals. P-values are given for each odds ratio and a significant result is indicated when the 95% confidence interval is completely greater than 0 or completely less than 0. (A) Escherichia-dominance had significantly higher odds of occurring in PFI < 6 months individuals (p-value = 0.024). (B) Antibiotics within one month was negatively associated with Lactobacillus-dominance, but not significantly (p-value = 0.051). There were no differences between medical/health/lifestyle variables in Prevotella (C) and highly diverse (D) vaginal microbiomes. Gardnerella-dominated communities were not included in this analysis due to small sample size (n = 3).

Approximately 24% (11 of 45) of patients in this study had Lactobacillus-dominated communities, which is significantly lower as compared to studies of similarly aged women without ovarian cancer (p-value = 0. 037) (Brotman et al., 2014a; Nené et al., 2019). Other studies have found Lactobacillus to be less abundant in Black and Hispanic women (Ravel et al., 2011). and our study consisted of 39 women who self-reported ethnicity as white, two self-reported as Black, and four self-reported as Native American (Table 1). Each of the Black and Native American women had a non-Lactobacillus dominated vaginal microbiome (Table S1) but ethnicity was not a statistically significant determinant of Lactobacillus-dominance (Fig 2B, log OR = −1.63, 95% CI [−4.62 to 1.30], p-value = 0.27). High microbial cell density, as gauged through qPCR with a standard curve generated from controls with known cell density, was positively correlated with vaginal Lactobacillus-dominance, although somewhat weakly (R2 = 0.278, Fig. S4). Consumption of antibiotics within the past month was associated with a lack of Lactobacillus-dominance (Fig. 2B); however, this relationship was not significant (log OR = −2.12, 95% CI [−5.54 to 0.84], p-value = 0.0515). Additionally, Lactobacillus abundance was uncommon in individuals likely to be estrogen negative (post-menopause and not on hormone replacement therapy) but this result was also not significant (log OR = −1.32, 95% CI [−2.83 to 0.33], p-value = 0.0744). Lactobacillus-dominance did not have a strong relationship with any of the other health or lifestyle factors we tested, including PFI length (Table S3, Fig. 2B). Only 20% (1 of 5) of patients with benign pathology had a Lactobacillus-dominated microbiome but small sample size prohibits statistical inference.

Previous studies have indicated that different Lactobacillus species in the vaginal microbiome may have different roles and differential influence on host biology (Lamont et al., 2011; Arumugam et al., 2011; Schloissnig et al., 2013). While we identified L. iners, L. brevis, L. mucosae, L. reuteri, L. zeae, and L. delbrueckii in the vaginal microbiome, 99% of the Lactobacillus reads in our study were either unclassified at the species level or mapped to L. iners (Table S4); ultimately there was not sufficient data to assess the relationship between most of these Lactobacillus species and clinical variables. Nevertheless, in individuals with high Lactobacillus abundance (n = 11), we found L. iners at significantly higher abundance in patients with either no gross residual disease or residual disease < 1 cm (n = 7), compared to patients with residual disease > 1 cm (n = 4, p-value = 0.0359, Fig. 3).

Figure 3 Lactobacillus iners dominates in small gross residual disease.

Lactobacillus reads not assigned to a species were significantly more abundant in individuals with Gross Residual >1 cm (p-value 0.02303). In individuals with Lactobacillus dominated vaginal microbiomes (n = 11), L. iners was at significantly higher relative abundance (p-value = 0.0359) in patients with no gross residual disease or residual disease under 1 cm.

We did not observe significant associations between either Prevotella and clinical variables (Table S3, Fig. 2C) or the high diversity community and clinical variables (Table S3, Fig. 2D). However, Prevotella was less common in those with residual disease > 1 cm and those with a history of hormonal disease (Fig. 2C), while a highly diverse vaginal microbiome was more common in women over 60 years old and in patients with a history of hormonal disorders, such as thyroid disease (Fig. 2D); but once again, none of these associations were statistically significant (Table S3, p-value > 0.05).

Gut microbiome

The gut microbiome was colonized by typical members of the gut microbiome at the phylum (Bacteroidetes, Firmicutes, Proteobacteria) and genus (Bacteroides, Akkermansia, Faecalibacterium, Ruminococcus, and Prevotella) levels (Fig. S5A–S5B). The PFI > 24 months and PFI < 6 months groups were not significantly different with respect to fecal unweighted and weighted UniFrac beta diversity distances as tested through PERMANOVA (Fig. 4, Fig. S6A–S6C); however, similar to the vaginal microbiome samples, there was a small subset of individuals (n = 9) with a unique microbiome signature. Patients with PFI < 6 months individuals had higher odds (log OR = 1.85, 95% CI [−0.85 to 4.497], p-value = 0.12) of being in this unique/outlier fecal microbiome group. These fecal outliers have significantly lower phylogenetic diversity compared to the other fecal samples (p-value = 0.001, Fig. 5A) and have increased abundance of genera belonging to the order Clostridiales (Lachnospira (Kruskal–Wallis p-value = 0.00039), unidentified Ruminococceae genus (Kruskal–Wallis p-value = 0.001337), and Subdoligranulum (Kruskal–Wallis p-value = 0.01121)) (Fig. S7). Of the six platinum-resistant patients in this outlier subset, two also had Escherichia-dominated vaginal communities. Eight of the nine patients in this subgroup reported consuming antibiotics within the past 6 months but this was not a significantly greater proportion of individuals with recent antibiotic consumption compared to individuals outside this fecal outlier subgroup (log OR = 0.35, 95% CI [−1.44 to 3.05], p-value = 0.48).

Figure 4 Unweighted UniFrac distances (PC1 and PC2) of fecal microbiomes from women with ovarian cancer.

Each shape represents a single sample and shapes clustering together have similar gut microbiome taxonomic composition. There was no significant difference in overall microbiome community structure between sample groups (PERMANOVA p-value > 0.05); however, there are nine samples (6 PFI < 6 months, 3 PFI > 24 months) that form an outlier group along PC1. These individuals are labelled.

Figure 5 Phylogenetic diversity in fecal microbiomes.

(A) Samples that formed the outlier group in Fig. 4 (n = 9) had lower phylogenetic diversity compared to the remainder of the gut microbiome samples (p-value = 0.0001). (B) Patients with PFI < 6 months had lower phylogenetic diversity than benign and platinum-sensitive patients but this was not a significant result (p-value = 0.18).

Overall, patients with platinum-resistant disease had lower fecal phylogenetic diversity compared to patients with platinum-sensitive disease, but the difference was not significant (Fig. 5B, p-value = 0.18). Regardless of PFI, patients with ovarian cancer had significantly higher relative abundance of Prevotella in the gut microbiome compared to benign individuals (p-value = 0.028, Fig. S8). Outside of the above-mentioned associations, there were no other significant associations between the gut microbiome and platinum sensitivity or other health/lifestyle variables.

Discussion

One of the major findings of this study is the inverse relationship between a Lactobacillus-dominant vaginal microbiome and ovarian cancer. In our study, fewer women than expected (24.4%) have Lactobacillus-dominated vaginal communities compared to similarly aged, healthy women (47.2%) from other studies (p-value = 0.037) (Brotman et al., 2014a; Nené et al., 2019). Lactobacillus dominance is not found in any of the Black (n = 2) or Native American (n = 4) women in our study but only eleven of the 39 white women have Lactobacillus-dominated vaginal microbiomes; therefore, ethnicity was not a driving factor in Lactobacillus abundance in this study (p-value = 0.27). These results suggest the possibility that the low abundance of Lactobacillus may be indicative of a broader relationship between ovarian cancer and the vaginal microbiome. This finding corroborates a previous study that also observed a reduced frequency of the Lactobacillus-dominated vaginal microbiome in women with ovarian cancer, particularly in women under 50 (Nené et al., 2019). A partial explanation may be that Lactobacillus-dominance, while typically viewed as a healthy state in the female genital tract, is a non-resilient ecology prone to disruption by variable factors, including changes in glycogen availability (Mirmonsef et al., 2014; Mirmonsef et al., 2015), antibiotic exposure (Melkumyan et al., 2015), and shifts in hormone abundance induced during stress responses (Witkin & Linhares, 2017). While we observed no statistical relationships between Lactobacillus dominance and platinum-sensitivity or other lifestyle/medical variables, there was a nearly significant decrease in Lactobacillus-dominance associated with taking antibiotics within the past month (p-value = 0.0515), as well as a positive association between Lactobacillus-dominance and microbial cell density (R2 = 0.278).

Lactobacillus maintains a low pH in the vaginal environment by producing lactic acid as a byproduct of glycogen metabolism and this low pH inhibits growth of pro-inflammatory bacteria (Champer et al., 2018; Chase et al., 2015; Mirmonsef et al., 2016). Vaginal Lactobacillus may protect from gynecological cancers by inhibiting pro-inflammatory bacteria, such as those implicated in pelvic inflammatory disease, and by reducing inflammatory cytokines IL-1β and IL-6 (Hemalatha et al., 2012). The low Lactobacillus levels we observed may be related to glycogen availability—nearly 75% of women in this study were post-menopause, and vaginally produced glycogen is known to decrease after menopause; likewise, chemotherapy can inhibit ovarian estrogen production and result in lower glycogen levels (Brotman et al., 2014a; Mirmonsef et al., 2016; Bachmann & Nevadunsky, 2000; Muhleisen & Herbst-Kralovetz, 2016). More importantly, many women in this study have had at least one ovary surgically removed during initial cancer treatment. Ovary removal leads to decreased estrogen production and thus, a likely decrease in vaginal glycogen levels; however, more research is needed to fully explain the ovarian-estrogen-glycogen dynamic (Mirmonsef et al., 2014). Glycogen abundance may also help explain the relationship between vaginal microbiome cell density and Lactobacillus, as widely available glycogen may encourage a densely colonized Lactobacillus vaginal community due to high nutrient availability (Muhleisen & Herbst-Kralovetz, 2016). Hormone replacement therapy (HRT) can be used to replace estrogen production that is stopped after menopause, and four women (two pre-menopause and two post-menopause) were on HRT. Women on HRT or pre-menopause were more likely to have Lactobacillus-dominated communities, but this was not significant (p-value = 0.0744). While we did not document glycogen levels in our study, the positive relationship between likely estrogen presence and Lactobacillus abundance lends credence to the idea that the low proportion of women with Lactobacillus abundance in our study is due to low estrogen, and thus glycogen. The retrospective nature of our study means that we were unable to assess Lactobacillus levels in women with ovarian cancer before they progressed to stage III/IV, or prior to chemotherapy. Therefore, we could not investigate anti-gynecological cancer properties of vaginal Lactobacillus; nevertheless, by comparing patients in this study to similarly aged women without ovarian cancer, we present further evidence that low Lactobacillus levels are more common in women with ovarian cancer (p-value = 0.037).

The presence, and size, of residual disease is strongly correlated with decreased survivability in ovarian cancer (Chang et al., 2013) and therefore our finding that L. iners was at significantly higher abundance (p = 0.0359) in patients with either no gross residual disease or residual disease < 1 cm after treatment may point to L. iners as a potential path toward a biomarker. L. iners is a common vaginal bacterium (Ravel et al., 2011; Lewis, Bernstein & Aral, 2017; Younes et al., 2017) but its role in health and disease is sometimes contradictory (Champer et al., 2018); L. iners has been found at high relative abundance in low-grade squamous intraepithelial lesions in the cervix but at low abundance in high-grade squamous intraepithelial lesions (Xu et al., 2020). Yet, another study found L. iners at high abundance in women with normal cytology when compared to women with squamous intraepithelial lesions (Audirac-Chalifour et al., 2016). Other studies have found L. iners to be positively associated with cervical cancer (Oh et al., 2015; Seo et al., 2016) but L. iners is also linked to clearance of HPV (Brotman et al., 2014b), which is a causative agent of cervical cancer. Further research is necessary to better understand the role of L. iners in gynecological cancers in general, and in the potential inhibition of gross residual disease in ovarian cancer.

Lactobacillus is known to inhibit colonization and growth of Escherichia in the vaginal microbiome (Delley et al., 2015; Gupta et al., 1998). The low abundance of Lactobacillus found in our study may present ample opportunity for typically low abundance vaginal bacteria, such as Escherichia, to thrive and proliferate in the absence of competition. Overgrowth of Escherichia only occurred in 17.8% of patients in our study, yet 75% of those patients were in the PFI < 6 months group; this finding was statistically significant (p-value = 0.024). Put another way, 35.3% of patients with PFI < 6 months showed greater than 20% relative abundance of Escherichia, as compared to 4.34% of PFI > 24 months and only one of 5 benign cases. The explanation for why Escherichia was significantly more common in patients with platinum-resistant tumors is unclear, and because this study was retrospective, we were unable to track Escherichia abundance before we knew each patient’s platinum-sensitivity. One possible pathway is via interactions between the microbiome, immune system, and how platinum-based chemotherapies induce cancer cell death. Platinum chemotherapies partially rely on ROS produced by host myeloid cells (Iida et al., 2013). Microbes strongly influence immune system function, and hence, alteration in ROS production may be more common in Escherichia-dominant vaginal microbiomes, which may render platinum-based chemotherapies less effective, leading to platinum-resistance. Vaginal Escherichia may also cause an increased inflammatory response, such as during pelvic inflammatory disease (Heinonen & Miettinen, 1994), and promote cancerous growth, resulting in a shortened PFI. The effect of Escherichia on platinum-sensitivity warrants further investigation.

We also observed differences in the gut microbiome of women with platinum-resistant tumors, compared to benign and platinum-sensitive. Similar to the prevalence of Escherichia-dominant vaginal microbiomes, 35.3% of patients with PFI < 6 months were fecal beta-diversity outliers compared to the remainder of the fecal samples, while only 13.0% of PFI > 24 and none of the benign cases fell into this cluster (p-value = 0.12). Genera belonging to the Clostridiales order (Subdoligranulum (p-value = 0.01121) and Lachnospira (p-value = 0.00039)) were at higher abundance in this group of outlier samples. Subdoligranulum, has been found at high abundance in the stool of individuals with gastrointestinal neoplasms (Youssef et al., 2018) and has a positive association with blood-based markers of inflammation (De Groot et al., 2017). Lachnospira has been found to be positively correlated with a plant-based diet (Vacca et al., 2020) and at high abundance in healthy controls in a study investigating chronic kidney disease (Lun et al., 2019); yet, Lachnospira is also found at high abundance in women with metabolic disorder and obesity (Vacca et al., 2020). Additionally, we note lower phylogenetic diversity in the gut microbiome of platinum-resistant tumors compared to both platinum-sensitive and benign tumors (p-value = 0.18). Chemotherapy is a well-documented driver of decreased gut microbiome alpha diversity (Montassier et al., 2015; Hakim et al., 2018; Alexander et al., 2017); however, we did not observe a shift in alpha diversity with time since last cycle of chemotherapy. High gut microbiome alpha diversity is typically associated with improved human health (Menni et al., 2017; Turnbaugh et al., 2009), although recent studies have started to question this paradigm (Reese & Dunn, 2018). Nevertheless, the unique gut microbiome beta diversity profile in a subset of PFI < 6 months, and the decreased alpha diversity in the full PFI < 6 months population indicates that there may be long term relationship between platinum-resistance and the gut microbiome.

Prevotella is enriched in both the platinum-resistant and platinum-sensitive study groups, compared to the benign group (p-value = 0.028). Prevotella is typically only found at high abundance in the gut microbiomes of non-industrial, traditional populations (Ley, 2016) and it is associated with consumption of a plant-rich diet (Ley, 2016). Yet, some strains of Prevotella are found in the gut microbiomes of industrial populations and are linked to pro-inflammatory states in the gut microbiome (Larsen, 2017; Scher et al., 2013). Similar to decreased Lactobacillus in the vaginal microbiome of patients with ovarian cancer, the relatively high abundance of Prevotella in the gut microbiome of women with ovarian cancer indicates a notable shift in microbial composition. Studies with larger control groups are necessary to address this relationship.

Conclusions

Our results demonstrate an association between the vaginal and gut microbiomes and platinum-sensitivity in women with ovarian cancer. Escherichia-dominant vaginal communities are significantly more likely to be present in patients with platinum-resistant tumors but the explanatory mechanism for this relationship is currently unclear. Lab contamination and/or collection methodology does not appear to play a role in vaginal Escherichia abundance, which indicates that finding Escherichia at high relative abundance in patients with PFI < 6 months is a biological trend.

We also observed shared vaginal and gut microbiome profiles in women with ovarian cancer, with decreased dominance of Lactobacillus and increased relative abundance of Prevotella, respectively, regardless of platinum-sensitivity. These results suggest shifts in microbiome composition that are related to the ovarian cancer disease state, which may possibly be related to chemotherapy, but the retrospective nature of our study does not allow us to distinguish the exact mechanism of action.

Our results call for deeper investigation into the relationship between the vaginal and gut microbiomes and ovarian cancer. A future avenue for research is a prospective, longitudinal study that tracks how the vaginal and gut microbiomes change throughout the course of ovarian cancer therapy, with an aim to disentangle how Escherichia-abundance impacts response to chemotherapy. Similarly, a study tracking Lactobacillus abundance in an aged-matched, lifestyle-matched cohort of women with and without ovarian cancer may provide insights into how microbial risk factors impact occurrence and outcomes of ovarian cancer. This work must also investigate why high L. iners abundance is found nearly exclusively in cases with < 1 cm or no gross residual disease, while other Lactobacillus species are found in cases with > 1 cm residual disease. Finally, ovarian cancer microbiome research also presents an opportunity for microbial metagenomics and metabolomics to provide a fuller picture of the vaginal and gut microbiome ecosystems in health and disease.

Supplemental Information

Supplemental Information 1 Flowchart of sample collection scheme.

Flowchart of sample collection for individuals in different study categories. Green circles represent treatment and yellow diamonds represent periods of sample collection. Individuals with benign tumors (n = 5) had samples collected immediately (1) and received no further treatment. Those with Stage III/IV epithelial cancer went through 6-8 cycles of platinum chemotherapy. Individuals enrolled in the study had samples collected as soon as they were identified as PFI < 6 months (2) or PFI > 24 months (3). Individuals were also enrolled in the study if they had previously been identified as PFI < 6 months or PFI > 24 months (prior to the start of the study) and samples were collected during treatment (Baldwin et al., 2012; Raja, Chopra & Ledermann, 2012) or during surveillance (Dasari & Tchounwou, 2014).

Click here for additional data file.

Supplemental Information 2 Weighted UniFrac beta diversity of all vaginal microbiome samples.

Each colored label is a sample from that patient number and colored lines point to the exact location for that label. Black lines connect samples from the same individual. Samples originating from the same individual had similar taxonomic composition. This informed our decision to combine data from the three vaginal samples per individual into a single sample per individual. Numbers within boxes represent sample ID.

Click here for additional data file.

Supplemental Information 3 Proportional contribution of most abundant phyla and genera in the vaginal microbiome in this study.

Stacked bar chart shows relative contribution of each phylum (A) and genus (B) to each individual’s vaginal microbiome and contributions from low abundance phyla and genera were combined into ‘Other’. Samples are organized along the x-axis by their relative abundance of the most dominant bacterium (Firmicutes/Lactobacillus). There was no clustering in producing this figure. Overall, the vaginal microbiome is dominated by the common vaginal bacteria: Firmicutes, Proteobacteria, and Bacteroidetes at the phylum level, and Lactobacillus, Prevotella, Escherichia, Gardnerella, and Streptococcus at the genus level.

Click here for additional data file.

Supplemental Information 4 Lactobacillus abundance has a positive association with log-transformed cell density in each sample.

Cubic polynomial fit to log cell density (x-axis) and reads mapping to Lactobacillus (y-axis). Log cell density is calculated from a qPCR standard curve created from standards with known concentration. The positive relationship (R2 = 0.2778) indicates that Lactobacillus abundance is related to microbial concentration in the vaginal environment.

Click here for additional data file.

Supplemental Information 5 Proportional contribution of most abundant phyla and genera in the gut microbiome in this study.

Stacked bar chart shows relative contribution of each phylum (A) and genus (B) to each individual’s gut microbiome and contributions from low abundance phyla and genera were combined into ‘Other’. Samples are organized along the x-axis by their relative abundance of the most dominant bacterium (Bacteroidetes/Bacteroides). There was no clustering in producing this figure. Overall, the gut microbiome is dominated by the common gut phyla/genera: Bacteroidetes, Firmicutes, and Proteobacteria, at the phylum level, and Bacteroides, Prevotella, and Akkermansia at the genus level.

Click here for additional data file.

Supplemental Information 6 Weighted UniFrac beta diversity for gut microbiome samples in this study.

Click here for additional data file.

Supplemental Information 7 Genera at high abundance in fecal outlier group.

Genera belonging to the Clostridiales order (Lachnospira, Ruminococacceae, Subdoligranulum) are at higher abundance in the gut microbiome beta diversity outlier group (Krukal-Wallis p-values = 0.00039, 0.001337, 0.01121). Parabacteroides is not significantly more abundant in the non-outlier group (Krukal-Wallis p-value = 0.1031).

Click here for additional data file.

Supplemental Information 8 Prevotella abundance in gut microbiome of ovarian cancer patients.

Patients with ovarian cancer (both platinum-sensitive and platinum-resistant) have higher levels of Prevotella in the gut microbiome compared to controls (p = 0.028).

Click here for additional data file.

Supplemental Information 9 Relationship between abundance of Escherichia in the vaginal and gut microbiomes.

Abundance of Escherichia in the gut microbiome (x-axis) and the vaginal microbiome (y-axis). There is a very weak positive relationship (R2 = 0.09) between Escherichia abundance in the gut and vaginal microbiome. This indicates vaginal Escherichia abundance is not due to fecal Escherichia abundance.

Click here for additional data file.

Supplemental Information 10 Full metadata for each individual.

Click here for additional data file.

Supplemental Information 11 Log Odds Ratios and p-values (metadata variables and study group).

Log-odds ratios and 95% confidence interval values, plus p-values, for every statistical comparison between the study groups (PFI< 6 and PFI > 24). Benign was excluded from statistical comparisons due to low sample size. Statistical tests for continuous variables (age and months since last platinum chemotherapy cycle) were conducted with a T test.

Click here for additional data file.

Supplemental Information 12 Log Odds Ratios and p-values (metadata variables and vaginal bacterial community).

Log-odds ratios and 95% confidence interval values, plus p-values, for every statistical comparison between the four dominant vaginal communities (Escherichia, Lactobacillus, Prevotella, and high diversity community) and different metadata categories.

Click here for additional data file.

Supplemental Information 13 Lactobacillus species abundance.

Number of reads mapping to each Lactobacillus species after rarefaction to 9000 reads, as well as percent of Lactobcaillus reads belonging to each species. More than 99% of reads that mapped to Lactobacillus were categorized as L. iners or unassigned to a species. The few individuals with reads mapping to any species besides L. iners prohibited analysis of how Lactobacillus species influences PFI status or any of the metadata collected in our study.

Click here for additional data file.

Supplemental Information 14 Sequencing data, Escherichia origin, and Code information.

Click here for additional data file.

The authors would like to thank Sarah Cooper and Cathy Birdsong for their work in coordinating patient recruitment and monitoring sample collection. The authors would also like to thank all the individuals who participated in this study.

Additional Information and Declarations

Competing Interests

Author Contributions

Human Ethics

DNA Deposition

Data Availability

The authors declare that they have no competing interests.

David Jacobson performed the experiments, analyzed the data, prepared figures and/or tables, authored or reviewed drafts of the paper, and approved the final draft.

Kathleen Moore conceived and designed the experiments, performed the experiments, authored or reviewed drafts of the paper, and approved the final draft.

Camille Gunderson conceived and designed the experiments, performed the experiments, authored or reviewed drafts of the paper, and approved the final draft.

Michelle Rowland performed the experiments, authored or reviewed drafts of the paper, and approved the final draft.

Rita Austin performed the experiments, authored or reviewed drafts of the paper, and approved the final draft.

Tanvi Prasad Honap analyzed the data, prepared figures and/or tables, authored or reviewed drafts of the paper, and approved the final draft.

Jiawu Xu performed the experiments, authored or reviewed drafts of the paper, and approved the final draft.

Christina Warinner conceived and designed the experiments, authored or reviewed drafts of the paper, and approved the final draft.

Krithivasan Sankaranarayanan conceived and designed the experiments, authored or reviewed drafts of the paper, and approved the final draft.

Cecil M. Lewis Jr conceived and designed the experiments, analyzed the data, prepared figures and/or tables, authored or reviewed drafts of the paper, and approved the final draft.

The following information was supplied relating to ethical approvals (i.e., approving body and any reference numbers):

The University of Oklahoma Health Sciences Center Institutional Review Board approved carry out of this study (Reference #6458).

The following information was supplied regarding the deposition of DNA sequences:

16S V4 rRNA sequences are available at NCBI SRA: SAMN16074939 to SAMN16075126.

The following information was supplied regarding data availability:

16S V4 rRNA sequences are available at NCBI: project number PRJNA662091.

Bioinformatic scripts for processing the 16S rRNA V4 data are available in the Supplemental File.

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
