# Peer review of "Shifts in gut and vaginal microbiomes are associated with cancer recurrence time in women with ovarian cancer"

_PeerJ, doi:10.7717/peerj.11574_

## Round 0.1 · original submission · Major Revisions

Both reviewers found your study study to be interesting and informative. But before we can make a decision on the publication of your manuscript, a number of concerns need to be addressed. These concerns are detailed in the reviews below. In particular, please ensure that the discussion of the figures and tables reflects the data presented. The statistical analysis needs to more complete (e.g. providing p-values) and presented along with the data. More detailed methods are needed, and the raw data and code also need to be made available.

·

Basic reporting

Figures and tables require clarification. As it is currently written, what is stated in the manuscript is not readily concluded from the figures and tables provided. Please see my specific comments to the authors.

Experimental design

The authors have a very interesting area of research, but I think their questions need clarifying. For example, the authors jump around from comparing PFI groupings to gross residual and then to Lactobacillus species content without explaining why these changes are made. In addition, the methods section would benefit from a study flow chart which describes how patients were included/excluded, when sampling occurred, and which treatments they received. Please see my comment to the authors for specific suggestions.

Validity of the findings

The authors need to report all p-values, not just the significant ones. I would recommend adding p-values to figures where appropriate and in supplemental tables if necessary.

There is no mention of depositing the sequence data in a public repository.

Additional comments

This is a retrospective study describing the vaginal and gut microbiomes of women undergoing treatment for ovarian cancer. The authors find that there are fewer women that have lactobacillus dominant vaginal microbiomes compared to previous studies. They also found a higher amount of Prevotella in the fecal microbiomes of these women.

Although I do think this study is interesting and important, I am not recommending publication at this time. There are some large wholes that need to be addressed before the conclusions of this study can be fairly reviewed. I describe specific issues below, but generally, this manuscript is missing methods, not all p-values are provided, and there are multiple issues with labelling graphs and figures. Often what is stated in the manuscript is not readily concluded from the figures and tables provided. However, I do think that most of the issues can be easily fixed. Once complete, I would recommend a second review process.

• Please be consistent in the groups that you are comparing. At some times you compare based on PFI, sometimes based on gross residual, and later on in the manuscript you compare based on Lactobacillus species. Make it clear why you are changing from one thing to another. Additionally, if you find a result that is significant in one comparison, please provide evidence that it is or is not significant in another comparison (e.g. Lactobacillus species vs PFI or gross residual).

• Please provide a table of *all* p-values that you have. A lot of p-values are stated in the text but are not listed in any figures or tables. I do not know how many comparisons were made if there was correction for type 2 error, or if anything else was significant that was not explicitly stated in the text.

• Table 1: Please change this to a summary statistics table.
o Three are 58 samples in that table, it is too many to be of any value. And all of this information (plus more) is provided in table S1. Please provide a summary statistics table. Or provide summary statistics for the compared groups (e.g. PFI>24 (n = 23), PFI<6 (n = 17), and benign (n = 5)). And then show what significant differences these groups have to one another (are they the same age, medical history, etc).
o Please make the formatting more readable. For example, the words are broken up in almost every column heading, and provide whole numbers (no decimals) for the ages.

• Please provide a study flow chart including the total patients screened, any excluded patients, when the samples were collected, and what treatments happened during the study. The methods section tries to explain that verbally, but it is unclear where the samples were taken. A flow diagram would clarify the process. For example, one part of the methods describes that the patients were recruited after they had been diagnosed as platinum resistant or super sensitive, but then later it describes that the patients were treated for 6-8 cycles. Is that total cycles or cycles post treatment? Were the samples collected before 6 cycles and after how many other cycles? Or were patients only treated if they had recurrence? A flow diagram will help with each of these questions. This diagram could also go into the supplemental data.

• Please describe the hormone status of your patient population in your summary statistics table and provide a comparison of the vaginal microbiomes to hormone status. As you state in the discussion, it is known that Lactobacillus levels are higher when estrogen is present, but you don’t describe the hormone state of these women during the study (+/- hysterectomy, menopause status, etc). Are all of these women estrogen negative? If so, that would go a long way to explaining the low level of Lactobacillus dominant vaginal profiles.

• Can you comment on the specific species of Lactobacillus you found in the vagina. L. brevis, L. mucosae, and L. zaea, are very uncommon species in healthy reproductive aged women or in healthy post-menopausal women. Is there anything known about these species and the vaginal microbiome? You have a nice description of L. iners in your discussion and describe that this species has both positive and negative associations. However, you do not touch on L. rueteri, and L. delbrueckii which are common in women’s health probiotics. I noticed in Supp Table 1 that you did collect information on probiotic use. Did you find any correlation between the detection of these species and probiotic use? Did you check which probiotics these women had taken (oral vs vaginal, brand or specific species?)

• In line 336 you state that L. iners is significantly higher abundance in patients with no or small residual disease. Do you do that same comparison with Lactobacillus genus as a whole? Also please fix the typo in the species name in this line.

Minor corrections:
• Please provide a figure description for each of the supplemental figures.

• Fig S1A-C:
- Instead of showing this as 3 2-D PCA plots, try visualizing this data as a 3D PCA plot with the axes of PC1, PC2, and PC3.
- In line 206-207 you state that the different vaginal sites showed similar beta-diversity. However, it is difficult to confirm this from figure S1A-C. In a 3D graph, potentially connect the related samples with a lines (making a triangle of samples). Or potentially show this in some other way.
- Line 208 – how did you group these 3 samples together? Did you take an average of each OTU? Please specify in the methods.

•Fig S2A-B:
- Please label panel A or B as vaginal microbiome at phylum and genus levels
- Please include the phylogenetic tree used to generate this figure. Why do the E. coli dominant samples in Panel A (ocm088, ocm106, ocm003) not cluster together? If these samples are not grouped by phylogenetic similarity please explain why and by what method they are grouped.

• Fig 1:
- Please explain the color coding labelled “Abundance (thousands)”. What does green, orange and purple indicate. The methods section mentioned that these samples were rarefied to 9000 reads, so are you trying to indicate the percentage of 9K that that particular genus represents?
- Please also include the phylogenetic trees that provide the sample clustering.
- Line228 – you point to this figure when describing which PFI groups the E. coli dominant profiles belonged to. However, the PFI groups are not listed in this figure.

• Fig S3:
- Again, this figure would benefit from a written description. I’m unclear what this bar chart represents. Is this the average profile of all the benign samples, PFI< 6, or PFI>24 samples? If so, please describe how you merged the samples in your methods section.
- In line 223, you say that 5 of the 6 Eschericia dominant communities IDed belonged to the PFI<6 months group, and you reference this figure. But I do not get any data representing how many patients (N of 5 or N of 20) are in each group. Please be sure that what you state in your manuscript has direct evidence that can be quickly understood by studying the corresponding figure.
- I think it would be nice to have this figure be more like Fig S2 with each individual patient’s profile next to each other. But grouped by PFI grouping. Alternatively you could modify figure S2 to include all of that information. Under each sample indicate which PFI group and which CST it belongs to.

• Fig 2:
- This figure has more demographic information that what is shown in Table 1. For example, you list the odds ratio for hormone disorder, antibiotics use, and Vit B and D supplementation. Yet these numbers do not occur in the summary statistics. It would be helpful to have that information in the summary statistics table, particularly if it is re-organized to be grouped by PFI grouping.
- The P-values comparing these odds ratios are not given. Please provide a table including all comparisons.

•Supplementary text: Thank you for providing a concise and rigorous argument as to why the E. coli was not an environmental contaminant.

• Line 238:
- How did you get this p-value? What groups are you comparing here and how are you comparing to a previously published study?

• Line 255:
- You specifically state that you found 6 different lactobacillus species in your study, but up to this point I have not seen anything that was classified to the species level. Please provide data showing the lactobacillus species distribution across your study.

• Line 262:
- This entire paragraph is still describing the odds ratios in figure 2 and should be moved up ahead of the paragraph describing the lactobacillus species distribution.

• Figure 3:
- Why the focus suddenly switch from PFI to gross residual disease?
- Was this comparison done for all Lactobacillus species and all demographic characteristics?
- Was there any difference in Lactobacillus species and PFI groups?
- Please italicize genus and species names in the figure description.

• Figure S5A-B:
- Again, please label with the phylogenetic trees used to generate the order of these bar charts. And explain the difference between panels A and B. It would also be helpful if there was some indication of the PFI grouping and vaginal microbiome CST.

• Figure S6A-C:
- Again, try presenting this data as a single 3D graph with lines or clouds indicating PFI groupings.

• Line 283:
- Again, where did you get this p-value from? Please provide a table of p-values from all comparisons done. If you report a single significant p-value but no others, then I have no way of assessing if there were other p-values that were not significant or maybe some that were but were not reported.

• Line 285:
- Please clarify that there was lower *fecal* phylogenetic diversity.

Reviewer 2 ·

Basic reporting

It doesn’t appear that the data or code were made available with the paper. The authors should make the data and code available, especially to the reviewers.

As a minor point, something may have gone wrong with the submission of figure 1. I was unable to read any legend or labels. It would be helpful if the authors updated this figure.

Experimental design

At line 206-210, it was interesting to see that the samples from the different vaginal sites were pooled before downstream analysis. Was there a statistical or other reason why the samples needed to be pooled? Are the authors worried about loosing signal by pooling the samples, even though they were compositionally similar by beta-diversity measurement? It would be helpful if the authors clarified the reasoning for this method in the text.

The data in figure S4 does not look linear, so it leaves the reader to wonder if this was the most appropriate method. Did the authors consider something like a cubic polynomial regression approach? If the linear model really is the best fit, it would be helpful for the authors to add some clarifying text indicating that other approaches were considered, but linear was most appropriate.

Validity of the findings

I appreciate that the authors wanted to highlight some visible differences despite their lack of statistical significance, but this made the paper confusing to read when the p-values were not cited, especially in the discussion section. For the sake of clarity, I recommend that the authors add p-values for each piece of comparative language, which mostly applies to the discussion section (for example, line 335 - 338). This would make it clearer to the reader and avoid the need to keep turning back to the results section to check the statistics values.

---

## Round 0.2 · accepted · Accept

I am pleased to be able to accept your manuscript for publication. Reviewer 1 did provide some minor suggestion which should be able to be handled during the publishing process.

·

Basic reporting

No comment

Experimental design

No comment

Validity of the findings

I have some very minor editorial suggstions:

The new figure 1 looks great
For figure 3 legend, please provide the p-value for the Gross Residual >1cm, even if it is not significant.
Please go through and double check proper italicization for species names. Particularly in the figure legends.
There is a small typo on Line 349: the word environment, the r is a superscript

Additional comments

I'm pleased with the changes the authors made.

Reviewer 2 ·

Basic reporting

No comment.

Experimental design

No comment.

Validity of the findings

No comment.

Additional comments

The authors adequately addressed the reviewer comments. The manuscript has improved and the conclusions are appropriate.